# Opposite regulation of inhibition by adult-born granule cells during implicit versus explicit olfactory learning

Nathalie Mandairon[1]*, Nicola Kuczewski[1], Florence Kermen[1], Jérémy Forest[1], Maellie Midroit[1], Marion Richard[1], Marc Thevenet[1], Joelle Sacquet[1], Christiane Linster[2,3], Anne Didier[1]

[1]Lyon Neuroscience Research Center, Neuroplasticity and Neuropathology of Olfactory Perception Team, CNRS UMR 5292, INSERM U1028, Université de Lyon, Lyon, France; [2]Computational Physiology Lab, Cornell University, Ithaca, United States; [3]Department of Neurobiology and Behavior, Cornell University, Ithaca, United States

**Abstract** Both passive exposure and active learning through reinforcement enhance fine sensory discrimination abilities. In the olfactory system, this enhancement is thought to occur partially through the integration of adult-born inhibitory interneurons resulting in a refinement of the representation of overlapping odorants. Here, we identify in mice a novel and unexpected dissociation between passive and active learning at the level of adult-born granule cells. Specifically, while both passive and active learning processes augment neurogenesis, adult-born cells differ in their morphology, functional coupling and thus their impact on olfactory bulb output. Morphological analysis, optogenetic stimulation of adult-born neurons and mitral cell recordings revealed that passive learning induces increased inhibitory action by adult-born neurons, probably resulting in more sparse and thus less overlapping odor representations. Conversely, after active learning inhibitory action is found to be diminished due to reduced connectivity. In this case, strengthened odor response might underlie enhanced discriminability.
DOI: https://doi.org/10.7554/eLife.34976.001

*For correspondence:
nathalie.mandairon@cnrs.fr

Competing interests: The authors declare that no competing interests exist.

## Introduction

Brain representations of the environment constantly evolve through learning mediated by different plasticity mechanisms. Several lines of evidence suggested that adult neurogenesis is a plasticity mechanism mediating changes in representations of sensory information (*Lledo and Valley, 2016*). Experience-dependent survival of adult-born neurons has received recent attention as a mechanism to modulate pattern separation and perceptual discrimination in the hippocampus and olfactory bulb (OB) (*Sahay et al., 2011*). In the OB, current hypotheses about the mechanism underlying this enhanced discrimination capability focus on increased inhibitory processes mediated by GABAergic adult-born neurons (*Moreno et al., 2009*; *Alonso et al., 2012*), with more integrating cells delivering more inhibition, leading to sparser odor representations. The data presented here challenge the current hypothesis by showing that the same rate of adult-born cell survival in the OB can enhance fine olfactory discrimination by either increasing or decreasing OB output sparseness. In the olfactory system, both passive (implicit perceptual learning in response to repeated exposure) and active (explicit associative learning in response to reinforcement) learning can improve discrimination between similar odorants (*Mandairon et al., 2006a*; *Moreno et al., 2009*). Both forms of learning have been shown to modulate neural activity in the OB (*Buonviso et al., 1998*; *Kay and Laurent, 1999*; *Doucette et al., 2011*), and to increase survival of inhibitory adult-born interneurons

(*Moreno et al., 2009*; *Sultan et al., 2010*; *Mandairon et al., 2011*; *Sultan et al., 2011*). However, exactly how adult-born neurons shape the output of the OB to support enhanced discrimination in these two paradigms is currently unknown. To address this question, we focused on how the number of integrated adult-born cells and/or the synaptic contacts established by adult-born neurons in the network could account for experience-induced changes in perception (*Moreno et al., 2009*; *Daroles et al., 2016*).

The data reported here, combining morphological analysis and optogenetic stimulation of adult-born neurons with mitral cells activity recording, indicate that both forms of learning similarly enhanced the number of adult-born neurons formed in the OB. However, these adult-born neurons showed higher density of dendritic spines compared to controls in implicit learning only, leading to stronger inhibition of mitral cells. In contrast, in explicit learning, adult-born neurons formed weaker connections to mitral cells, resulting in disinhibition of mitral cells in learning conditions compared to control. These experiments indicate that a same number of adult-born inhibitory neurons are able to positively or negatively modulate inhibition in the OB, depending on the synaptic integration mode and dictated by the behavioral significance of the odorant.

## Results

### Implicit learning increased adult-born granule cell survival and spine density

To induce implicit learning, animals were exposed to two odorants, which are not spontaneously discriminated: +limonene (+lim) and - limonene (-lim). These were placed in two tea balls in the home cage for one hour per day over 10 days as previously described (*Moreno et al., 2009*, *2012*) (Enriched animals; *Figure 1A*). Controls were exposed to empty tea balls (Non-enriched animals). We used a habituation/cross habituation test to assess the effect of the enrichment on discrimination between +lim and –lim (see methods) (*Moreno et al., 2009*, *2012*). Significant habituation was observed in both the control (n = 20; Friedman test p<0.0001) and enriched groups (n = 25; Friedman test p<0.0001). Enriched animals discriminated +lim and -lim at the end of the enrichment procedure as evidenced by increased investigation times between the last habituation trial (OHab4) and the test trial (OTest) (Wilcoxon test for discrimination p=0.004) whereas the controls did not (Wilcoxon test for discrimination p=0.74) (*Figure 1B* and *Figure 1—source data 1*).

We used Bromodeoxy-uridine (BrdU) incorporation to quantify adult-born neuron survival and lentiviral transduction to follow their morphological integration into the network (GFP encoding lentivirus was injected 8 days before enrichment to allow for adult-born neurons to migrate from the subventricular zone to the OB; *Figure 1A*). As expected, implicit learning increased adult-born granule cell survival (p=0.042, parametric Anova followed by Bonferroni post hoc test; Non-Enriched n = 6 and Enriched n = 4, see methods for global statistics strategy), (*Table 1*; *Figure 1C* and *Figure 1—source data 1*; *Figure 1—figure supplement 1A*) as well as their responsiveness to the learned odors as assessed by BrdU/Zif268 co-expression (p=0.0078, parametric Anova followed by Bonferroni post hoc tests; Non-Enriched n = 7 and Enriched n = 6), (*Table 1*; *Figure 1D* and *Figure 1—source data 1*; *Figure 1—figure supplement 1B*) (*Moreno et al., 2009*). Spine density of labeled adult-born neurons was analyzed on the apical (site of interactions with mitral/tufted (M/T) cells) and the basal (site of input of centrifugal fibers) dendrites of adult-born granule cells (*Figure 1E*). Spine density increased after implicit learning on both the apical (p=0.0003, Kruskal-Wallis Anova followed by FDR-corrected permutation tests; Non-Enriched: 48 segments, four mice and Enriched: 30 segments, four mice) (*Table 1*; *Figure 1F* and *Figure 1—source data 1*) and basal dendritic domains (p=0.0015, Kruskal-Wallis Anova followed by FDR-corrected permutation tests; Non-Enriched: 59 segments, seven mice and Enriched: 48 segments, seven mice), (*Table 1*; *Figure 1G* and *Figure 1—source data 1*). In addition, 5 to 6 weeks after enrichment, once the enriched animals had forgotten the task, the density of BrdU-positive cells and the spine density on adult-born cells had returned to control values (*Figure 1—figure supplement 2*).

### Inhibition on mitral cells is increased after implicit learning

We then analyzed how the observed neurogenic changes affect OB output by investigating the overall level of inhibition of the M/T cells. Spontaneous inhibitory postsynaptic current (sIPSC) frequency

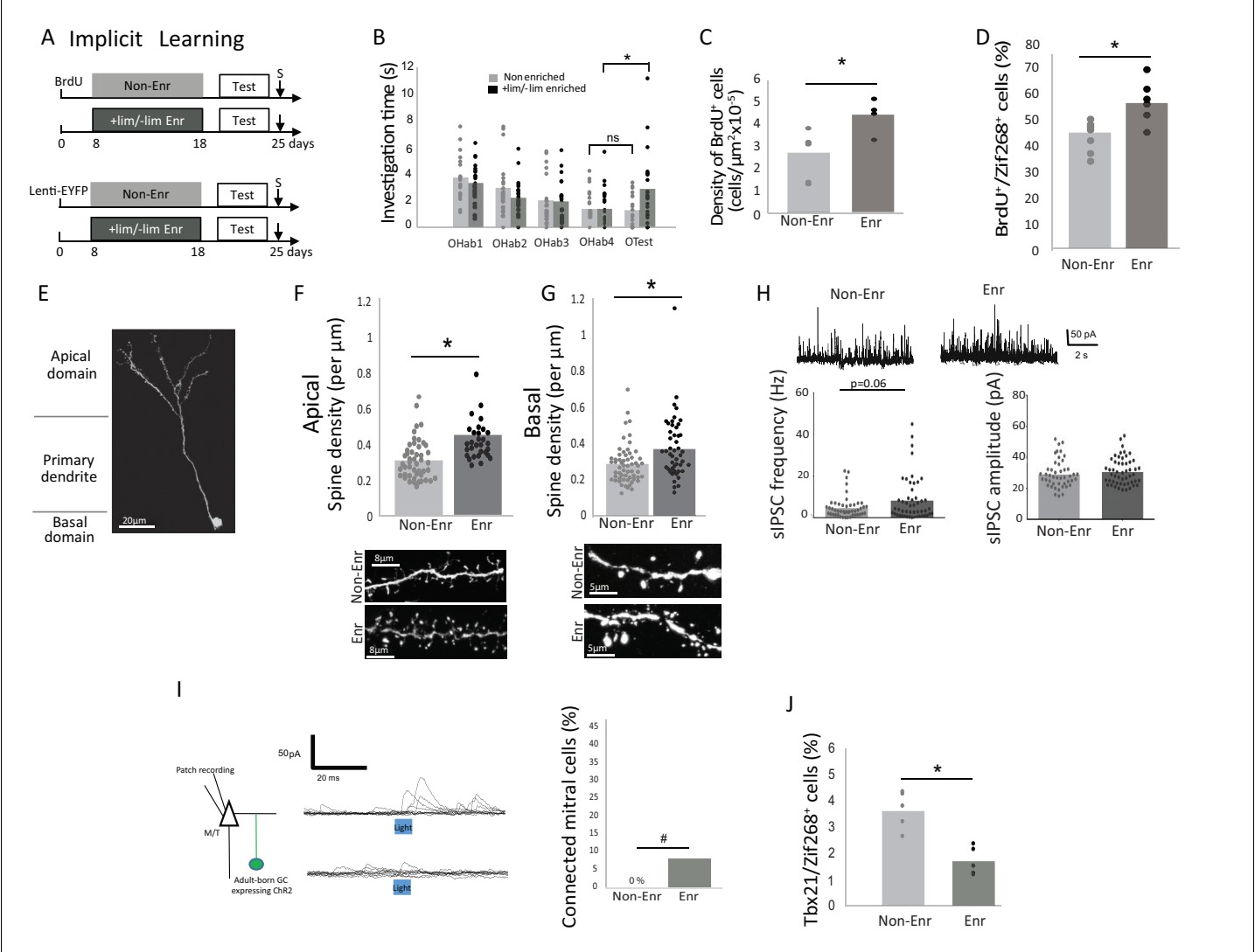

**Figure 1.** Behavioral and neural effects of implicit learning. (**A**) Experimental design for implicit learning (S: Sacrifice; Test: Habituation/Cross Habituation task). (**B**) Behavior. Habituation/cross habituation task indicated that +lim and -lim are not discriminated in control non-enriched group. In contrast, enrichment allows discrimination as observed by the significant increase of investigation time between the last habituation trial (OHab4) and the presentation of the second odorant of the pair (Otest). (**C**) Adult-born cell (BrdU-positive cell) density is increased after implicit learning. (**D**) The percentage of odor-responsive adult-born cells (expressing Zif268) is increased after implicit learning. (**E**) Spine density of adult-born neuron transduced by Lenti hSyn ChR2EYFP is analyzed in the apical and basal domains. (**F**) Spine density in the apical domain is increased after implicit learning. (**G**) Spine density in the basal domain is increased after implicit learning. (**H**) Representative traces of sIPSC recorded on mitral cells for Enr and Non-Enr animals (up). sIPSC frequency is increased after implicit learning while no modification is observed for sIPSC amplitude (down). (**I**) Left, experimental design for studying the connectivity of adult-born neurons on M/T cells. Middle, example of the effect of optogenic stimulation of adult-born neurons on M/T cells IPSC (top connected M/T cell, bottom unconnected M/T cell, superposition of 10 traces). Right, percentage of M/T cells exhibiting a significant response to light stimulation of adult-born neurons. (**J**) The percentage of M/T cells (Tbx21positive) expressing Zif268 is decreased after implicit learning. *:p<0.05; #p=0.07.

DOI: https://doi.org/10.7554/eLife.34976.003

The following source data and figure supplements are available for figure 1:

**Source data 1.** Raw Data *Figure 1*
DOI: https://doi.org/10.7554/eLife.34976.009
**Figure supplement 1.** Adult-born neuron survival and responsiveness to learned odorant.
DOI: https://doi.org/10.7554/eLife.34976.004
**Figure supplement 2.** Long-term delay after implicit learning.
DOI: https://doi.org/10.7554/eLife.34976.005
**Figure supplement 3.** Effect of light stimulation of adult-born neuron on mitral cell activity.

*Figure 1 continued*

DOI: https://doi.org/10.7554/eLife.34976.006

**Figure supplement 4.** The biophysical properties of adult-born neurons.

DOI: https://doi.org/10.7554/eLife.34976.007

**Figure supplement 5.** Example of Tbx21/Zif268-positive mitral cell.

DOI: https://doi.org/10.7554/eLife.34976.008

recorded on M/T cells showed a trend toward increase after implicit learning (p=0.06, Kruskal-Wallis Anova followed by FDR-corrected permutation test, Enriched, n = 50 and Non-Enriched, n = 48) (for detailed statistics see *Table 1*), (*Figure 1H* and *Figure 1—source data 1*) with no change in amplitude (*Table 1*, Enriched n = 48 and Non-Enriched, n = 46). To isolate the specific role of adult-born granule cells in the inhibition of M/T cells among the global granule cell population, we then optogenically activated channelrhodopsin expressing adult-born granule cells in OB slices (adult-born granule cells were transduced with Lenti-hSyn-ChR2EYFP 25- 30 days before testing) and recorded the evoked inhibitory post-synaptic current (eIPSC) in M/T cells (Enriched n = 27 cells from three mice; Non-Enriched n = 25 cells from three mice) (*Figure 1I*, left). Data analysis indicated an overall effect of light stimulation of granule cells on M/T cells eIPSC frequency (light effect, parametric Anova, $F_{(1,198)}=4.37$, p=0.037, paired t-tests for light versus pre light: p=0.0025 in the enriched group, p=0.2 in the non-enriched group) (*Figure 1—figure supplement 3A*). Besides, eIPSC frequency was higher in enriched compared to Non-enriched animals (learning effect, parametric ANOVA, $F_{(1,198)}=4.47$, p=0.03, t-test for difference between Enriched and non-enriched under light stimulation, p=0.04) (*Figure 1—figure supplement 3A*). No change in eIPSC amplitude was observed (parametric ANOVA, light effect $F_{(1,160)}=2.66$, p=0.1, learning effect $F_{(1,160)}=1.72$, p=0.19) (*Figure 1—figure supplement 3B*). To identify the individual M/T cells actually responding

**Table 1.** Summary of statistical comparisons described in the text.

For normal data, Anova followed by parametric Bonferroni *post hoc* test were used. For data that did not reach normality, Kruskall-Wallis Anova followed by FDR-corrected permutation tests were used. *p<0.05; **p<0.001; ***p<0.0001 and =: not different

| | Cond vs Enr | PC vs Non-Enr | PC vs Enr | PC vs Cond | Cond vs Non-Enr | Enr vs Non-Enr |
|---|---|---|---|---|---|---|
| **BrdU** Anova $F_{(3,18)}=6.63$, p=0.003 | Cond = Enr (p=0.99) | PC = Non-Enr (p=0.99) | PC < Enr * (p=0.024) | PC < Cond * (p=0.025) | Cond > Non-Enr * (p=0.048) | Enr > Non-Enr * (p=0.042) |
| **BrdU/Zif268** Anova $F_{(3,23)}=9.25$, p=0.0003 | Cond = Enr (p=0.99) | PC = Non-Enr (p=0.99) | PC < Enr * (p=0.006) | PC < Cond * (p=0.005) | Cond > Non-Enr * (p=0.0073) | Enr > Non-Enr * (p=0.0078) |
| **Tbx21/Zif268** Anova $F_{(3,15)}=7.33$, p=0.002 | Cond > Enr * (p=0.015) | PC = Non-Enr (p=0.053) | PC = Enr (p=0.99) | PC < Cond * (p=0.044) | Cond = Non-Enr (p=0.99) | Enr < Non-Enr * (p=0.02) |
| **Spine density Apical** Kruskal-Wallis $H(3, N = 240)=44.55$ p<0.0001 | Cond < Enr *** (p=0.0003) | PC = Non-Enr (p=0.59) | PC < Enr * (p=0.001) | PC > Cond * (p=0.00255) | Cond < Non-Enr * (p=0.0132) | Enr > Non-Enr ** (p=0.0003) |
| **Spine density Basal** Kruskal-Wallis $H(3, N = 187)=20.15$ p<0.0001 | Cond = Enr (p=0.36) | PC > Non-Enr ** (p=0.0006) | PC = Enr (p=0.36) | PC = Cond (p=0.09) | Cond > Non-Enr * (p=0.042) | Enr > Non-Enr * (p=0.0015) |
| **sIPSCs (Frequency)** Kruskal-Wallis $H(3, N = 177)=10.68$ p=0.013 | Cond = Enr (p=0.55) | PC > Non-Enr ** (p=0.0006) | PC = Enr (p=0.13) | PC > Cond (p=0.06) | Cond = Non-Enr (p=0.13) | Enr > Non-Enr (p=0.06) |
| **sIPSCs (Amplitude)** Kruskal-Wallis $H(3, N = 175)=21.30$ p<0.0001 | Cond < Enr * (p=0.0048) | PC = Non-Enr (p=0.48) | PC = Enr (p=0.7) | PC > Cond * (p=0.015) | Cond < Non-Enr * (p=0.018) | Enr = Non-Enr (p=0.48) |

DOI: https://doi.org/10.7554/eLife.34976.002

to adult-born granule cell stimulation, we performed a statistical comparison of the pre- and post-light IPSC occurrence across repeated stimulations (see Materials and methods). Results indicated that the percentage of light-responding M/T cells is marginally increased in enriched compared to non-enriched animals (unilateral Chi squared, p=0.07; *Figure 1I* right and *Figure 1—source data 1*). Additionally, we showed that the biophysical properties of adult-born granule cells (resting potential, membrane resistance, membrane capacitance and input/output curves) were unaffected by learning (*Figure 1—figure supplement 4*). Thus, the results from these experiments suggest that increased structural and/or functional connectivity is responsible for increased granule-to-M/T cell inhibition in enriched versus non-enriched groups, rather than any modification of the intrinsic properties of adult-born granule cells. To assess the functional outcome of this increased inhibition, we investigated in vivo M/T cell responsiveness to the learned odorant (+lim). Enriched and non-enriched animals were exposed to +lim 7 days after learning, and the percentage of odor-activated M/T cells (co-expressing the specific M/T cell marker Tbx1 (*Imamura et al., 2011*; *Mitsui et al., 2011*) and immediate early gene Zif268) was quantified (*Figure 1—figure supplement 5*). We found that implicit learning decreased the number of odor-activated M/T cells compared to non-enriched controls (587 ± 29 M/T cells counted per animal; Enr, n = 5 and Non-Enr, n = 4; Bonferroni-corrected test p=0.02; *Table 1*, *Figure 1J* and *Figure 1—source data 1*), resulting in a sparser representation of learned odorants at the OB output. This finding is highly consistent with the increased inhibition of M/T cells observed in slice recordings. Altogether, in the context of implicit learning, adult-born granule cells provide more inhibitory contacts on M/T cells, leading to increased sparseness of M/T odor responses. As a consequence, these findings strongly support the hypothesis that a decreased overlap between learned odor representations mediates the observed increase in odor discrimination (*Chu et al., 2016*).

## Explicit learning increased adult-born granule cell survival but reduced spine density

Theoretically, discrimination could also be achieved through increasing the resolution of the odor representations (*Aimone et al., 2011*), in other words through increasing the response to the learned odorant (*Doucette and Restrepo, 2008*). Indeed, adding information to the representation of the stimulus could ultimately help discrimination, and we hypothesized that such a mechanism could be involved in associative learning where the odor is reinforced by a positive reward. We thus trained mice on a two odorized hole-board apparatus to associate +lim with a food reward while -lim was not reinforced (Conditioned group, Cond) (*Mandairon et al., 2006a*). Control animals were exposed to the same odorants randomly associated with the reward (pseudo-conditioned group) (*Mandairon et al., 2006a*; *Sultan et al., 2010*) (*Figure 2A*). Training took place over 5 days (four trials/day) and was evidenced by an increase of correct choices in the conditioned (n = 20; Friedman test day effect p=0.03) but not in the pseudo-conditioned group (PC) (n = 20; Friedman test day effect p=0.57; *Figure 2B* and *Figure 2—source data 1*). Using the habituation/cross habituation test, we confirmed that conditioned but not pseudo-conditioned animals discriminated +lim from –lim as they did with implicit learning (*Figure 2—figure supplement 1*). As with implicit learning, explicit learning induced an increase in the survival of adult-born granule cells (n = 6/group; Bonferroni-corrected test p=0.025; *Table 1*; *Figure 2C* and *Figure 2—source data 1*) as well as an increase in their responsiveness to the learned odor (+lim), as assessed by BrdU/Zif268 co-expression (n = 7/group; Bonferroni-corrected test p=0.005; *Figure 2D* and *Figure 2—source data 1*). However, in contrast to implicit learning, analysis of the fine morphology of adult-born neuron dendrites showed a decrease in spine density on the apical dendrites after conditioning compared to pseudo-conditioning (Cond 106 segments, four mice and PC 56 segments, five mice; non-parametric corrected test p=0.0025; *Figure 2E* and *Figure 2—source data 1*), while no such change was observed in the basal domain (Cond: 37 segments, four mice and PC: 43 segments, four mice; non-parametric corrected test p=0.09; *Figure 2F* and *Figure 2—source data 1*). 42 days after conditioning, when discrimination had returned to control levels (*Figure 2—figure supplement 2A*), the spine density had also returned to control values (*Figure 2—figure supplement 2B*). This indicated that the morphological changes of adult-born neurons paralleled discrimination performance.

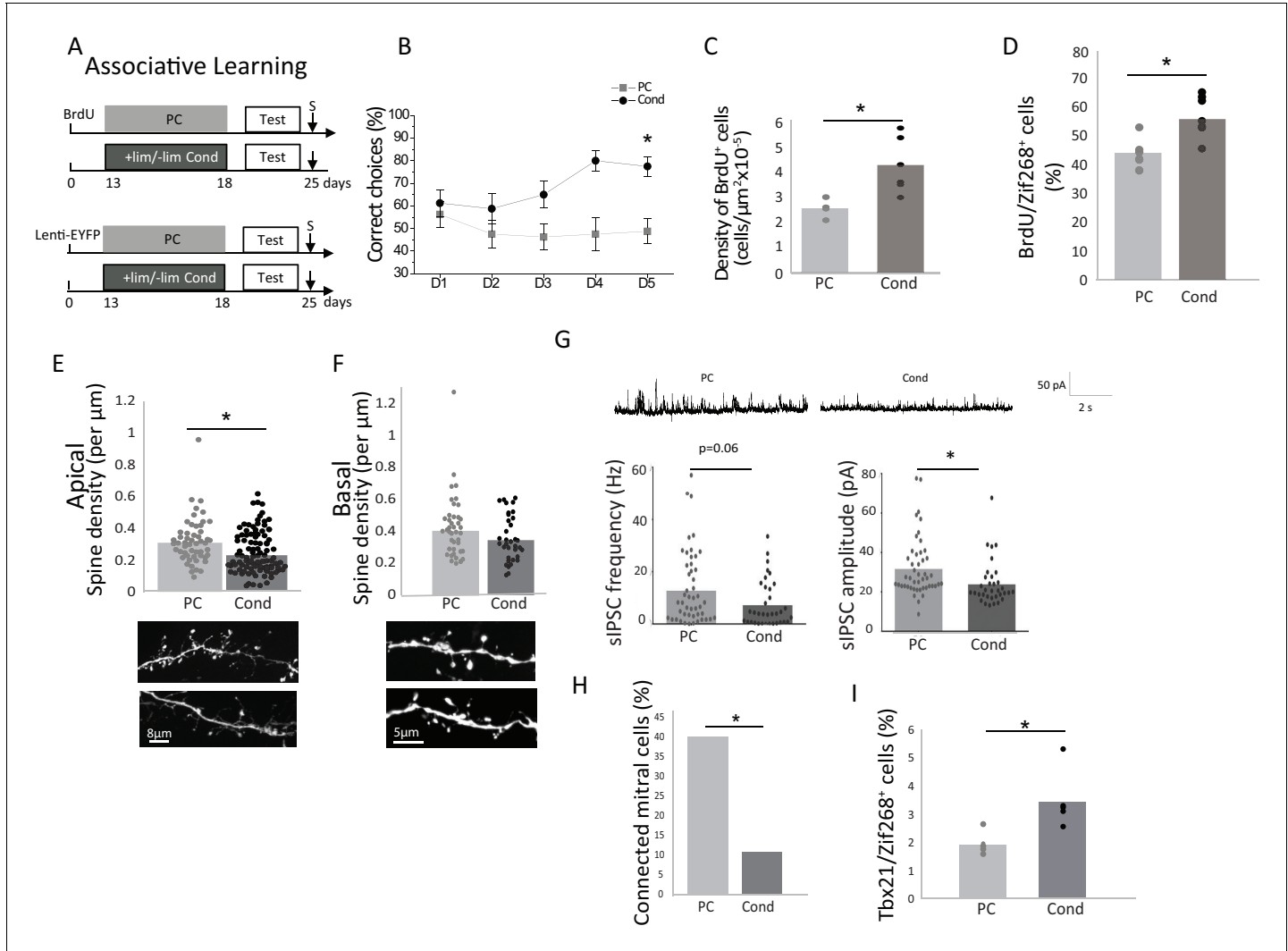

**Figure 2.** Behavioral and neural effects of explicit learning. (**A**) Experimental design for explicit learning (S: Sacrifice, Test:Habituation/Cross Habituation Task). (**B**) From D1 to D5 of training, the percentage of correct choices increased in conditioned (Cond) but not pseudo-conditioned animals (PC) indicating that learning occurred only in the conditioned animals (**C**) Adult-born cell (BrdU-positive cell) density is increased after explicit learning (**D**) The percentage of odor-responsive adult-born cells (expressing Zif268) is increased after explicit learning. (**E**). Spine density of the apical domain decreased after explicit learning. (**F**) Spine density of the basal domain is unchanged after explicit learning. (**G**) Representative traces of sIPSC for Cond and PC (up). sIPSC frequency and amplitude are decreased after explicit learning (down). (**H**) Percentage of mitral cells exhibiting a significant response to light stimulation of adult-born granule cells. (**I**) The percentage of mitral cells (Tbx21) expressing Zif268 is higher in Cond versus PC animals. *p<0.05; the data are expressed as mean values ± SEM.

DOI: https://doi.org/10.7554/eLife.34976.010

The following source data and figure supplements are available for figure 2:

**Source data 1.** Raw Data *Figure 2*.
DOI: https://doi.org/10.7554/eLife.34976.015
**Figure supplement 1.** Improvement in discrimination after explicit learning.
DOI: https://doi.org/10.7554/eLife.34976.011
**Figure supplement 2.** Long-term delay after explicit learning A.
DOI: https://doi.org/10.7554/eLife.34976.012
**Figure supplement 3.** Effect of light stimulation of adult-born neuron on mitral cell activity.
DOI: https://doi.org/10.7554/eLife.34976.013
**Figure supplement 4.** The biophysical properties of adult-born neurons.
DOI: https://doi.org/10.7554/eLife.34976.014

## Inhibition on mitral cells is decreased after explicit learning

We further analyzed the impact of adult-born neurons on M/T cell activity by recording sIPSCs from M/T cells in OB slices. sIPSCs amplitude was decreased (p=0.015, Kruskal-Wallis Anova followed by FDR-corrected permutation test; *Table 1*) while their frequency was marginally affected (p=0.06, Kruskal-Wallis Anova followed by FDR-corrected permutation test; *Table 1*) (*Figure 2G* and *Figure 2—source data 1*) suggesting a global reduction of M/T cell inhibition in the OB in conditioned versus pseudo-conditioned animals. To analyze specifically the impact of adult-born granule cells on M/T cells, we recorded IPSCs on M/T cells in response to light stimulation of channelrhodopsin-expressing adult-born granule cells. The light stimulation of adult-born granule cells induced an increase in M/T cell eIPSC frequency in both conditioned and pseudo-conditioned animals compared to pre-light (light effect, parametric Anova, $F_{(1,198)}=4.37$, p=0.037, paired t-tests for light versus pre light, p=0.015 in the pseudo-conditioned group, n = 46 cells from five mice; p=0.001 in the conditioned group, n = 34 cells from three mice) (*Figure 2—figure supplement 3A*) with no change in amplitude (*Figure 2—figure supplement 3B*). Besides, eIPSC frequency was lower in conditioned compared to pseudo-conditioned animals (learning effect, parametric Anova, $F_{(1,198)}=4.47$, p=0.03, t-test for difference between conditioned and pseudo-conditioned groups under light stimulation, p=0.0015; Pseudo-conditioned n = 22 cells from two mice; Conditioned n = 28 cells from two mice) (*Figure 2—figure supplement 3A*). Analysis of individual cell responses showed the percentage of M/T cells responding to light stimulation of adult-born granule cells by an increased eIPSC frequency was lower in conditioning compared to pseudo-conditioning (Unilateral Chi squared test, p=0.007), (*Figure 2H* and *Figure 2—source data 1*). The biophysical properties of adult-born granule cells (resting potential, membrane resistance, membrane capacitance and input/output curves) were not affected by explicit learning (*Figure 2—figure supplement 4*). As a consequence, we concluded that inhibitory inputs to M/T cells were decreased after explicit learning. The functional outcome of this decreased inhibition was assessed using Tbx21/Zif268 co-expression in mitral cells (540 ± 20 mitral cells counted per animal; n = 5 animals/group), and showed a significant increase in mitral cell responsiveness to the learned odor, (Bonferroni-corrected test p=0.04; *Figure 2I* and *Figure 2—source data 1*) suggesting that in this case, odor representations at the OB output were not rendered sparser by learning.

## Comparison between implicit and explicit learning

Using a two-factor Anova, we found that explicit and implicit learnings similarly increased adult-born survival (learning versus controls $F_{(1,18)}=19.72$, p=0.0003), regardless of the type of learning (implicit versus explicit $F_{(1, 18)}=0.21$, p=0.64 and no interaction $F_{(1,18)}=0.006$, p 0.9) (*Figure 1C* and *Figure 2C*). The same was true for adult-born cells responsiveness to the learned odor (learning effect $F_{(1,23)}=27.70$, p<0.0001, no effect of the type of learning $F_{(1, 23)}=0.23$, p=0.87, no interaction $F_{(1,23)}=0.0001$, p=0.98) (*Figure 1D* and *Figure 2D*). However, implicit learning resulted in stronger inhibition on M/T cells than did explicit learning as shown by lower sIPSC amplitude in conditioned versus enriched groups (p=0.0048, FDR-corrected permutation test, *Table 1*). In addition, adult-born granule cells bear more spines in enriched compared to conditioned groups in their apical domain which interacts with mitral cells (p=0.0003, FDR-corrected permutation test, *Table 1*, *Figure 1F,G*; *Figure 2E,F*). In line with this, M/T cells displayed lower responsiveness to the learned odorant in enriched compared to conditioned animals (p=0.015, Bonferroni *post hoc* test, Tbx21/Zif268, *Table 1*, *Figure 1J* and *Figure 2I*).

Interestingly, when comparing the controls for each learning group (pseudo-conditioned versus non-enriched) (*Table 1*), they appeared to differ. More precisely, sIPSC frequencies were higher in the pseudo-conditioned compared to the non-enriched animals (p=0.0006, FDR-corrected permutation test). Consistent with this, the number of odor-activated M/T cells tended to be smaller in the pseudo-conditioned than the non-enriched animals (p=0.053 Bonferroni *post hoc* test, *Table 1*). These differences could be explained by the fact that the pseudo-conditioned animals, in contrast to the non-enriched animals were exposed to the odorants throughout the pseudo-conditioning procedure.

Finally, we observed that the pseudo-conditioned animals shared cellular similarities with enriched animals (similar sIPSC frequency, percentage of odor-activated M/T cells and basal spine density) (*Table 1*) despite the fact that they do not show behavioral discrimination.

## Discussion

The findings reported here reveal that enhanced odor discrimination following implicit and explicit learning is achieved through different mechanisms. While the number of integrated adult-born granule cells was similar in both forms of learning, they differed in the synaptic integration mode of adult-born neurons and their effect on M/T cell responses to odor. Implicit learning increased spine density on adult-born granule cells (apical and basal dendritic domains), in agreement with previous studies (*Daroles et al., 2016*; *Zhang et al., 2016*) and increased inhibition of mitral cells, consistent with reduced number of mitral cells responding to the learned odorant. Increased number of spine in the basal domain is suggestive of an enhanced connectivity between inputs from centrifugal projections and adult-born granule cells, possibly leading to more global excitation of adult-born granule cells (*Moreno et al., 2012*; *Lepousez et al., 2014*). More apical spines increase feedback inhibition between M/T and granule cells increasing local inhibition. These data suggest that in response to implicit learning, structural plasticity of adult-born cells mediates an increased feedback and central inhibition on mitral cells to support perceptual discrimination of odorants. This view is strongly supported by our previous report of enhanced paired-pulse inhibition in the OB after implicit learning (*Moreno et al., 2009*), and of the loss of learning upon blockade of neurogenesis (*Moreno et al., 2009*). In addition to increased spine density, the increase in the number of adult-born cells after implicit learning is also likely contributing to the enhancement of inhibition on mitral cells.

In contrast to the effects of implicit learning, a decrease in spine density in the apical domain of adult-born neurons is accompanied by a decrease in sIPCS amplitude in mitral cells after explicit learning. In addition, an overall increase rather than a decrease of mitral cells activation was observed in response to the learned odorant compared to pseudo-conditioned animals. Reduced synaptic contacts on the apical dendrites of adult born neurons reduce local feedback inhibition leading to an enhanced response of M/T cells to the learned odorants.

To summarize, the effects of implicit and explicit learning on M/T odor responses are opposite: an overall sparser response to the learned odor after implicit learning and an overall increased response to the conditioned odor after explicit learning, while similar numbers of adult-born neurons are present. Because new adult-born granule cells replace older ones (*Imayoshi et al., 2008*), replacing pre-existing granule cells by new ones with fewer synaptic contacts with mitral cells (in conditioned animals) would result in a global pool of granule cells delivering less local inhibition in response to the conditioned odor. In contrast, replacing granule cells by new cells making more local and global synaptic contacts with mitral cells (enriched animals) would result in a shift toward more inhibition in the network. This is consistent with the experimental observations. Computational modeling suggested that reinforced inhibition reduces the overlap between similar odor representations at the level of mitral cells (*Mandairon et al., 2006b*). As an alternative mechanism leading to behavioral discrimination, decreased inhibition could lead to enhanced representations of a conditioned odor only, improving the resolution of the representation of behaviorally significant stimuli (*Aimone et al., 2011*). This proposed mechanism that could be at play in explicit learning is consistent with the associative coding features of the OB (*Doucette et al., 2011*; *Fletcher, 2012*). While these evidences favor a prominent role of adult born neurons in shaping mitral cell activity in learning, the role of pre-existing interneurons in mitral cell response plasticity remains to be investigated.

In conclusion, the data presented here challenge current hypotheses by showing that the same rate of adult-born cell survival in the OB can enhance fine discrimination by either increasing or decreasing OB output, thus revealing a new facet of the adaptability of the network provided by adult neurogenesis.

## Materials and methods

### Animals

Adult male C56Bl6/J mice (Charles River, L'arbresles, France) aged 8 weeks at the beginning of the experiments were used in this experiment. They were housed in groups of five in standard laboratory cages with water and food *ad libitum* (except during the explicit learning) and were kept on a 12-hr light/dark cycle (at constant temperature of 22°C). All behavioral training was conducted in the afternoon (14:00-17:00). Experiments were done following procedures in accordance with the European

Community Council Directive of 22nd September 2010 (2010/63/UE) and the National Ethics Committee (Agreement DR2013-48 (vM)). Every effort was made to minimize suffering.

## Implicit (perceptual) learning

### Experimental design

After a 10-day enrichment period, the mice were tested for their ability to discriminate between +limonene (+lim) and – limonene (- lim) using an olfactory cross-habituation test (see below). This group was compared to a non-enriched control group. To label the adult-born cells, the mice were injected with BrdU or GFP expressing lentivirus 8 days before the beginning of the enrichment period. They were sacrificed 25 (*Figures 1* and *2*) or 60 (*Figure 1—figure supplement 2* and *Figure 2—figure supplement 2*) days after these injections.

### Odor enrichment and control

Odor enrichment consisted of exposure to + and - limonene (purity >97%; Sigma-Aldrich Corp., Sr. Louis, MO, USA) for 1 hr per day over 10 consecutive days. The odors were presented simultaneously on two separate swabs containing 100 µl of pure odor placed in two separate tea balls hanging from the cover of the animal's cage. The non-enriched control mice were housed under the same conditions except that the two tea balls were left empty.

### Olfactory habituation/cross habituation test

We used a cross-habituation test to assess discrimination. Briefly, the task assesses the degree to which mice are able to discriminate between odorants by habituating them to an odorant (OHab) and measuring their cross-habituation to a second odorant (OTest). If the test odorant is not discriminated from the habituation one, it will not elicit an increased investigation time by the mouse. We presented each odor in a tea ball hanging on the cover of the cage containing 60 µl of the diluted odor (1 Pa) on filter paper (Whatman #1) for 50 s. There was a 5 min pause between presentations. Odors are renewed between each test. For each animal, each odorant of the pair was used alternatively as the habituation or test odorant. The amount of time that the mice investigated the odorant was recorded during all trials as previously described (*Moreno et al., 2009*, *2012*). We determined (1) if the investigation time decreased from OHab1 to OHab4 (habituation); and (2) if the investigation time during OTest was significantly higher than for OHab4 (discrimination). These behavioural experiments were conducted blind with regard to experimental group.

## Explicit (associative) learning

### Experimental design

The mice learned to discriminate +lim and - lim by training them to associate +lim with a food reward (see below). This group of mice were compared to a pseudo-conditioned control group. After a 5-day period of conditioning, the mice were tested for their ability to discriminate between +limonene (+lim) and – limonene (- lim) using an olfactory cross-habituation test (see above). They were injected with BrdU or with GFP lentivirus 13 days before training and sacrificed 25 (*Figure 1* and *Figure 2*) or 60 days after injection. During the olfactory learning experiments, water was continuously available, but the mice were deprived of food (~20% reduction of daily consummation, leading to a 10% reduction in body weight) for 5 days before the shaping session.

### Shaping

The mice were first trained to retrieve a reward (a small bit of sweetened cereal; Kellogg's, Battle Creek, MI, USA) by digging through the bedding. The mouse was put in the start area of the two hole-board apparatus and allowed to dig for 2 min. During the first few trials, the reward was placed on top of the bedding in one of the holes. After the mice successfully retrieved the reward several times, it was successively buried deeper and deeper in the bedding. Shaping was considered to be complete when a mouse could successfully retrieve a reward buried deep in the bedding.

## Conditioning

Conditioning consisted of 5 sessions (1/day) of 4 trials (2 min/trial, inter trial interval 15 min). For each trial, the mouse was placed in the start area and had to retrieve a reward now systematically associated with the +lim (20 µl of pure odorant). To avoid spatial learning, the rewarded dish was randomly placed in one of the two holes; the other hole contained – lim (20 µl of pure odorant) and no reward. In the pseudo-conditioned group, the reinforcement was randomly associated with either the +lim or the - lim. For each trial, correct choices (first nose poke in the odorized hole) were recorded as indicative of learning.

## Lentivirus injections

The mice were anesthetized by injecting a mixture of 50 mg/kg ketamine and 7.5 mg/kg xylazine (intraperitoneal) and then secured in a stereotactic instrument (Narishige Scientific Instruments, Tokyo, Japan). For the optogenetic experiments, they were injected bilaterally in the subventricular zone (AP +1 mm, ML ±1 mm, DV −2.3 mm) with a AAV5-hSyn-ChR2EYFP viral vector (150 nl per side; Addgene 26973P). This construct was a kindly donated by the Deisseroth laboratory and produced by the Penn Vector Core facility (titer 1,3 1013 UI/ml). For the neuronal morphology experiments, a Lenti-PGK-GFP (Addgene #12252; titer 2 $10^9$ UI/ml) viral vector was injected at the same coordinates (200 nl per site). All injections were performed at a rate of 100 nl/min using a programmable syringe controller (KD Scientific Inc. Holliston, USA).

## Histology

### Tissue preparation

Mice randomly taken from the behavioral groups were placed in a clean cage for 1 hr. To investigate immediate early gene expression in response to the learned odorant (+lim), the mice were presented with a tea ball containing 100 µl of pure odorant for 1 hr. One hour after the end of the odor stimulation, they were deeply anesthetized (pentobarbital, 0.2 ml/30 g) and killed by intracardiac perfusion of 50 ml of fixative (4% paraformaldehyde in phosphate buffer, pH 7.4). Their brains were removed, postfixed, cryoprotected in sucrose (20%), frozen rapidly, and then stored at −20°C before sectioning (40 µm for neuronal morphology and 14 µm for BrdU analysis) with a cryostat (Reichert-Jung, NuBlock, Germany).

### Image analysis and quantification

Morphological analysis of adult-born cells was performed after GFP immunohistochemisty (chicken GFP antibody, 1:1000, Anaspec TEBU, ref: 55423). Images were acquired on a Zeiss pseudo-confocal system. For analysis of spine density, images were taken blind to the identity of the experimental group with a 100x objective (lateral and z-axis resolutions were 60 and 200 nm, respectively). Dendritic processes and spines were then analyzed using NeuronStudio software (*Mandairon et al., 2006a*; *Rodriguez et al., 2008*). This software allows for automated detection of three-dimensional (3D) neuronal morphology (dendrites and spines) from confocal z-series stacks on a spatial scale. Because spine density assessment was challenging for the automated detection in our model, we counted them manually with the help of the 3D reconstruction. All morphology analyses were done blind with regard to the experimental group.

## Assessment of neurogenesis

### 5-Bromo-2-deoxyuridine (BrdU) administration

BrdU (Sigma-Aldrich) (50 mg/kg in saline, 3 times at 2 hr intervals) was injected 13 days before the behavioral training began.

### BrdU immunohistochemistry

The protocol has been previously described (*Mandairon et al., 2006a*). Brain sections were first incubated in Target Retrieval Solution (Dako, Trappes, France) for 20 min at 98°C. After cooling for 20 min, they were treated with 0.5% Triton X-100 (Sigma-Aldrich) in PBS for 30 min, then for 3 min with pepsin (0.43 U/ml in 0.1 N HCl, Sigma-Aldrich). Endogenous peroxidases were blocked with a solution of 3% $H_2O_2$ in 0.1 M PBS. Sections were then incubated for 90 min in 5% normal horse serum (Vector Laboratories, Burlingame, CA, USA), in 5% BSA (Sigma-Aldrich) and 0.125% Triton

X-100 to block nonspecific binding and were incubated overnight at 4°C in a mouse anti-BrdU primary antibody (1:100; Millipore; ref: MAB4072). Sections were then incubated in a horse biotinylated anti-mouse secondary antibody (1:200; Vector Laboratories) for 2 hr and processed with avidin-biotin-peroxidase complex (ABC Elite Kit, Vector Laboratories) for 30 min. Finally, sections were reacted in 0.05% 3,3-diaminobenzidine-tetra-hydrochloride (Sigma-Aldrich), 0.03% $NiCl_2$, and 0.03% $H_2O_2$ in Tris-HCl buffer (0.05 M, pH 7.6), dehydrated in graded ethanols, and cover-slipped in DPX.

### BrdU-positive cell quantification

All cell counts were conducted blind with regard to experimental group. Data were collected with the help of mapping software (Mercator Pro; Explora Nova, La Rochelle, France), coupled to a Zeiss microscope (Carl Zeiss, Oberkochen, Germany). BrdU-positive cells were counted in the granule cell layer of the OB on six sections distributed along the olfactory bulb (14 µm thick, 70 µm intervals). The number of positive cells was divided by the surface of the region of interest to yield the total densities of labeled cells (labeled profiles/µm$^2$). All BrdU-positive cell counts were done blind with regard to the experimental group.

### Double-labeling analysis

To determine the phenotype of BrdU-positive cells of the OB, BrdU/neuronal nuclei (NeuN) double-labeling was performed using a rat anti-BrdU (1:100; Abcys ref: Abc117-7513) and a mouse anti-NeuN (1:500, Millipore MAB337). For functional involvement of newborn neurons, Zif268/BrdU double-labeling was performed using a rabbit anti-Zif268 antibody (1:1000, Santa Cruz Biotechnology, Egr-1 SC:189). The appropriate secondary antibodies, coupled to Alexa 633 (Molecular Probes, Eugene, OR, USA) to reveal BrdU and Alexa 488 (Molecular Probes) to reveal other markers, were used. BrdU-positive cells were examined for co-labeling with NeuN and Zif268 (80–100 cells/animal, $n$ = 5 animals/group). The double-labeled cells were observed and analyzed by pseudo-confocal scanning microscopy using a Zeiss microscope equipped with an Apotome. All double-labeled cell counts were done blind with regard to the experimental group.

## Zif268-Tbx21 double labeling

The activation of mitral cell in response to the learned odorant (+lim) was assessed using co-labeling with Zif268 and Tbx21 (1:20,000; provided by Y. Yoshihara, RIKEN). The appropriate secondary antibodies (Molecular Probes, Eugene, OR, USA) were used. All Zif268-Tbx21 cell counts were done blind with regard to the experimental group.

## Brain slice electrophysiological experiments

For brain slice experiments, 20 additional mice were injected with lentivirus expressing GFP as described previously and submitted to implicit or explicit learning. Electrophysiological recording and data analysis were done blind with respect to experimental group.

The animals were anesthetized with intraperitoneal injection of 50 µl of ketamine (50 mg/ml) between 25 and 30 days after the lentivirus injections and killed by decapitation. The head was quickly immersed in ice-cold (2–4°C) carbogenized artificial cerebrospinal fluid (cACSF; composition: 125 mM NaCl, 4 mM KCl, 25 mM NaHCO$_3$, 0.5 mM CaCl$_2$, 1.25 mM NaH$_2$PO$_4$, 7 mM MgCl$_2$ and 5.5 mM glucose; pH = 7.4) oxygenated with 95% O$_2$/5% CO$_2$. The osmolarity was adjusted to 320 mOsm with sucrose. OBs were removed and cut into horizontal slices (400 µm thick) using a Leica VT1000s vibratome (Leica Biosystems, France). These were then incubated in a Gibb's chamber at 30 ± 1°C using an ACSF solution with a composition similar to the cACSF, except that the CaCl$_2$ and MgCl$_2$ concentrations were 2 mM and 1 mM, respectively. Slices were transferred to a recording chamber mounted on an upright microscope (Axioskop FS, Zeiss) and were continuously perfused with oxygenated ACSF (4 ml/min) at 30 ± 1°C. Neurons were visualized using a 40X objective (Zeiss Plan-APOCHROMAT) and a Hamamatsu 'Orca Flash 4.0' camera. Measurements were performed with an RK 400 amplifier (BioLogic, France). The data were acquired with a sampling frequency of 25 kHz on a PC-Pentium D computer using a 12-bit A/D-D/A converter (Digidata 1440A, Axon Instruments) and PClamp10 software (Axon Instruments). The junction potential was corrected offline. Patch-clamp configurations were achieved with borosilicate pipettes (o.d.: 1.5 mm; i.d.: 1.17 mm; Clark Electromedical Instruments). Membrane properties and firing input/output curves of adult

born neurons were recorded with the following intracellular solution: in mM (131 K-gluconate; 10 HEPES; 1 EGTA; 4 ATP-Na2+; 0.3 GTP-Na3; 1 MgCl2; 10 phosphocreatine). To record the spontaneous IPSCs (sIPSC) on mitral cells and those evoked by optogenetic stimulation of adult-born granule cells (see below) the recording pipette was filled with the following intracellular solution (in mM; 135 caesium gluconate, 10 KCl, 10 Hepes, 1 EGTA, 0.1 CaCl2, 2 MgATP and 0.4 GTP-Na3) adjusted with 1 M CsOH to pH 7.3. The ECl was −85 mV and both sIPSC and light evoked IPSCs were recorded as outward currents at holding potential of 0 mV.

## Optogenetic experiment

EYFP expression in adult-born granule cells was detected with excitation (470/40 nm) and emission (dichroic mirror: 495; 525/50 nm) band pass filters (Zeiss filter set 38 HE) 25–30 days after their birth. Optogenetic stimulation of adult-born cells was produced by 10 ms illumination by a blue LED (Dual Port OptoLED CAIRN, UK) with the excitation spectrum centered at 470 nm at an inter-stimulus interval of 10 s. For each recorded mitral cell the connectivity with adult-born granule cells was evaluated by statistically comparing the occurrence of IPSCs in the 50 ms preceding the light stimulation with the IPSC occurrence in the first 50 ms following the beginning of light stimulation, across repeated light stimulations (mean = 16 stimulations per cell, range 8–52). Pre-light and post-light IPSC frequency was calculated as follows: the total number of IPSC in the 50 ms preceding and following the beginning of light stimulation was divided by the number of repetitions and multiplied by 0.05 s.

## Statistics

All analyses were performed using Statistics. A Kolmogorov-Smirnov test was used to assess normality of measured parameters. For all parametric tests, homogeneity of variance was tested using Levene's test.

For each habituation/cross habituation task, a non-parametric Friedman Anova was performed to determine whether mice exhibited habituation, and a paired Wilcoxon test (comparing Ohab4 and Otest) to test the discrimination abilities. Discrimination was indicated by a significant increase in investigation time during the test trial. For conditioning task, a non-parametric Friedman Anova was used to assess the effect of training days on the percentage of correct choice.

Non-parametric Kruskal-Wallis Anova including the four experimental groups (Enr, Non-Enr, PC and Cond) followed by FDR-corrected non-parametric permutation tests based on 1000000 artificial groups, were used for sets of data that did not reach normality (*Table 1*).

For normally distributed data sets, parametric Anovas followed by Bonferroni-corrected *post hoc* tests were used (*Table 1*).

In addition, for eIPSC data, in order to take into account the factors 'light' (pre and post light), 'group of learning' (implicit versus explicit) and 'learning' (Cond or Enr versus PC or Non-Enr), we normalized the data ($\ln(x + 1)$) and performed a 3-factor Anova followed by an unilateral paired t-test for comparison between light and no light condition or an unpaired t-test for the comparison between learning and control condition. In addition, individual cell analysis of the effect of light was performed by comparing the occurrence of pre and post light eIPSC across repetitions of light stimulation by unilateral Chi squared tests.

No statistical methods were used to predetermine sample sizes, but our sample sizes were similar to those reported in previous publications. Data collection and animal assignation to the various experimental groups were randomized.

## Acknowledgements

This work was supported by the CNRS, Inserm, and Lyon 1 University. We would like to thank K Deisseroth for the gift of the channelrhodopsin construct, C Benetollo from the Neurogenetic and Optogenetic Platform of the CRNL for lentiviral production and G Froment, D Nègre and C Costa from the lentivector production facility/SFR BioSciences de Lyon (UMS3444/US8). We thank Y Yoshihara (Riken Brain Science Institute, Saitama Japan) for the gift of the Tbx21 antibody. We thank I Caillé for her comments and S Scotto-Lomasesse for her help with the morphological analysis. We would like to thank Y Chelminski, A Auguste and A Ferréol for helping during behavioral procedure and cell morphology analysis.

## Additional information

### Funding

| Funder | Author |
|---|---|
| Centre National de la Recherche Scientifique | Nathalie Mandairon |
| Université Claude Bernard Lyon 1 | Nathalie Mandairon |

The funders had no role in study design, data collection and interpretation, or the decision to submit the work for publication.

### Author contributions

Nathalie Mandairon, Conceptualization, Data curation, Formal analysis, Supervision, Investigation, Methodology, Writing—original draft, Writing—review and editing; Nicola Kuczewski, Data curation, Formal analysis, Writing—original draft; Florence Kermen, Data curation, Formal analysis, Writing—original draft, Writing—review and editing; Jérémy Forest, Maellie Midroit, Data curation, Formal analysis; Marion Richard, Methodology, Writing—original draft; Marc Thevenet, Formal analysis, Methodology; Joelle Sacquet, Data curation, Formal analysis, Methodology; Christiane Linster, Conceptualization, Writing—original draft; Anne Didier, Conceptualization, Formal analysis, Supervision, Writing—original draft, Writing—review and editing

### Author ORCIDs

Nathalie Mandairon http://orcid.org/0000-0001-7935-5215
Anne Didier http://orcid.org/0000-0003-3118-9961

### Ethics

Animal experimentation: Experiments were done following procedures in accordance with the European Community Council Directive of 22nd September 2010 (2010/63/UE) and the National Ethics Committee (Agreement DR2013-48 (vM)). Every effort was made to minimize suffering.

### Decision letter and Author response

Decision letter https://doi.org/10.7554/eLife.34976.018
Author response https://doi.org/10.7554/eLife.34976.019

## Additional files

### Supplementary files

• Transparent reporting form
DOI: https://doi.org/10.7554/eLife.34976.016

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
