## [Decision Letter]

Thank you for submitting your work entitled "Opposite regulation of inhibition by adult-born granule cells in response to implicit versus explicit olfactory learning" for consideration by *eLife*. Your article has been reviewed by three peer reviewers, and the evaluation has been overseen by a Reviewing Editor and a Senior Editor.

Summary:

Mandairon and colleagues examined the effects of two types of olfactory learning on granule cell neurogenesis, spine density of adult-born granule cells (GCs) and mitral cell (MC) activity and physiology. The learning paradigms compared are (1) presentation of two different odors [(+)- and (-)-limonene] in the home cage (1 hour/day, 10 days) ("implicit learning"), and (2) odor-reward associative learning using a food-digging task (explicit [associative] learning).

The data suggest that implicit learning increased inhibitory drive onto mitral cells (assessed based on the frequency of IPSC after a short pulse of ChR2 stimulation of adult-born GCs) and decreased odor-evoked activity of the mitral cells (as measured by immediate early gene expressions). These changes were accompanied by increased spine densities both at the apical and basal dendrites of GCs. In contrast, in explicit learning, the frequency of light-evoked IPSCs was decreased in MCs, and odor-evoked activity in MCs is increased. These changes were accompanied by a decrease in spine density at apical but not basal dendrites of GCs.

Although both implicit and explicit learning induced increased discriminability in the test phase as tested by the same assay (habituation paradigm), the authors found different morphological and physiological changes in GCs and MCs. All the reviewers agreed these are potentially very interesting findings. However, all pointed substantive concerns regarding the use of statistical methods and the robustness of the data. Specifically, some conclusions were made without correcting for multiple comparisons. Furthermore, statistical significance of some of the analyses appeared to be due to outlier points, raising the issue of robustness. After discussing these concerns, the reviewers thought that it is very likely that the authors need to perform additional experiments, or significantly change their conclusions (because some results might not be statistically significant after correcting for multiple comparisons). It was the consensus of the reviewers that addressing these issues will take more than 2-3 months. We therefore decided that we cannot consider the current manuscript further at least in the present form. If the authors improve their statistical arguments either by revising the manuscript to use more statistically sound analysis or by performing more experiments to increase the sample size, we are willing to consider a revised manuscript as a new submission.

Essential revisions:

1) All the reviewers raised their concerns regarding statistical analyses. Most importantly, the authors perform multiple pair-wise comparisons. In such cases, a significance criterion should be corrected for multiple comparisons (e.g. by using Bonferroni correction). When appropriate, a one-way ANOVA (analysis of variance) followed by post-hoc pair-wise comparisons is preferred. Many conclusions do not appear to hold after these corrections. This would mean that some conclusions need to be revised or the authors need to perform additional experiments to increase the sample size. More detailed criticisms and suggestions can be found in the reviewers' comments appended below.

2) Some of the statistical significance appear to be due to some outlier points in the data (e.g. reviewer 1, point #1; reviewer 2, point #2). The authors need to consider whether the results hold even removing these points. To make these results more reliable, we expect that a larger sample size is required to draw statistically-sound conclusions. The reviewers also thought that the conclusion that adult-born neurons are mediating the enhanced inhibition onto MCs requires more work:

3) How do authors control for number of cells expressing ChR2 virus or normalize counts to interpret the ChR2 stimulation data (number of spines vs. number of cells)? Are equivalent numbers of cells infected in both groups? The increase in survival induced by learning should bias the number of Chr2 expressing GCs in OB.

4) Following reversal of changes in apical spine density (but with extra adult-born GCs still present), is the effect of ChR2 stimulation on MCs lost?

5) The link between behavior (higher discrimination) and the changes at the level of neurogenesis, morphology and physiology remain largely unclear. We understand that this study is not designed to examine the mechanisms that connect these changes with behavior, but the reviewers found some of the discussions very loose. We would like the authors to state these links more carefully, so that the manuscript becomes more scholarly. For example, the authors mention "pattern separation" multiple times and discuss that increased or decreased inhibition may be causally related pattern separations, and in turn, increased discriminability. However, whether pattern separation causally underlies increased discrimination or whether changes in inhibitory tones can lead to pattern separations remain totally unknown (also see point #4 by reviewer 3). The present study as well as existing literature does not specify specific mechanisms underlying increased discriminability after implicit or explicit olfactory learning.

*Reviewer #1:*

In this manuscript, the authors describe bidirectional changes in the spine density of olfactory bulb adult-born granule cells (aGCs), that depend on learning paradigms which the authors have established previously.

The used paradigms appear subtle and physiological, since they rely on few odor exposures. Nevertheless, the changes can be observed in electrophysiological recordings in brain slices even though the relation of the investigated mitral cells to odor-activated glomeruli is not known. Thus, the phenomena might reflect a general response of an otherwise deprived olfactory system.

My general criticism is that there are counterintuitive results that would require stronger statistics to be firmly established. Therefore, the authors should increase sample size, try to sample cells more specifically (see below) and thoroughly discuss their results, possibly requiring a longer format.

1) Statistical significance appears to be an issue with some results – would unilateral naris occlusion be feasible, which might allow for pairwise comparisons across hemibulbs? Or a higher number of animals?

While the increased spine density (Figure 1) looks fine, all the other changes (Figure 1, Figure 2) show highly overlapping distributions, perhaps because some MCs in the sample might not have been involved in limonene-evoked bulbar activation. Would it be possible to preferentially record from bulbar parts with strong limonene activation? Or even better fluorescently label MC/GCs expressing IEGs and record from them?

There is a substantial amount of eIPSC recordings with high frequency in the 30-60 Hz range in PC condition (Figure 2). This range is not observed in Figure 1, even not in the enriched case. Were the high frequency responses recorded in slices from the same animal? Strikingly, the very same issue arises also for the sIPSC frequency and amplitude (Figure 2) – for both there are several data points in the high range for PC that are not observed in non-enr or enr conditions. If we now doubt the PC data for this reason, what about the statistical comparison between non-enr and cond? Unfortunately, this comparison is missing from Table 1 – by eye between Figure 1 and Figure 2 almost certainly there is no difference for the apical spine density. Which casts doubt on the central finding of reduced connectivity – at least anatomically.

2) What is the effect of the increase in number of adult born GCs also in the associative learning – how exactly can an addition of interneurons reduce inhibitory drive? As mentioned above, the decrease in spine numbers on the apical dendrite is barely significant. Nevertheless, there is a striking overall decrease in inhibitory drive of mitral cells, especially substantially reduced sIPSC frequency and amplitude (please label the examples from Figure 2 and Figure 1 top in the distributions shown below). If we believe these data (but see above), what is going on here? Synaptic plasticity – reduction in release probability and/or quantal size at the GC-MC contacts – also for preexisting GCs? Were the lost contacts located close to the MC somata and thus the efficiency is strongly reduced?

3) Missing of important reference: Zhang, Huang and Hu, (2016): increased spine density on GC dendrites following odor enrichment in *Xenopus* tadpoles.

*Reviewer #2:*

This report by Mandairon and colleagues is a significant new look on the function of adult born GABAergic neurons in the olfactory bulb. The community as a whole thinks in two ways about the types of learning that occur with passive exposure and conditioning. On the one hand, we all know that these are different types of learning and on the other we often refer to them as if they are the same (under the umbrella term of Learning). Similarly, for learning-based effects on the survival and integration of new granule cells in the OB. This paper shows that the ability to discriminate hard-to-discriminate odors after the two types of learning may depend in opposite fashion on neurogenesis. For passive learning, it appears that adult-born GC connections with MCs are strengthened in the external plexiform layer (at the reciprocal synapse). The authors conclude that this may be related to the effect known as sparsening. For active learning, the effects are opposite. The adult born GC effects on MCs are decreased. The argument is rigorous and rational. My only major concern regards the many pairwise statistical tests (see below). The results reported here, if improved statistical methods confirm them, may constitute one of the most significant findings in the field of olfactory learning as it relates to neurogenesis.

1) There are many more implied comparisons than reported in Table 1. With 4 groups of mice, this makes 6 total comparisons for each factor (cond vs. PC, cond vs. enr, cond vs. non-enr, PC vs enr, PC vs non-enr, enr vs non-enr). With the multiple comparisons within each factor, the threshold for significance should be 0.00833. There is a pretty good argument that all of the factors should be combined in a single analysis, which then multiplies the comparisons and lowers the threshold value, or at least that like-kind factors be grouped (spine density apical/basal as one group, e/sIPSP frequency as another). This would make the p threshold smaller by a factor of 2 at least. Please provide justification in the methods for not performing the multiple comparisons adjustment or do the analysis with the corrected p thresholds. In addition to lowering the threshold, a more conservative analysis might provide additional insight.

2) This is not a disagreement but rather a different interpretation. It is the adult born cells that show the effects reported (and assuming that the earlier born cells do not). In the passive condition, the result is that new GCs inhibit MCs more at the apical dendrites where primary processing happens. Also, we know that these new cells are relatively specific to the enriched odors (previous work by the first and second to last authors). Is it possible that these new connections serve to help the mice ignore the conditioned stimuli and that the enhanced discrimination ability is a side-effect of this now-active ignorance? On the other hand, in the case of active learning there are more cells born and integrated (perhaps associated with the learned odors). Because these cells survive, one assumes that the inputs from higher order areas are strong. The signal is amplified, so one might expect that the now meaning-based odor perception is accomplished in the AON or PC or both, rather than in the OB.

*Reviewer #3:*

Mandairon and colleagues examined the effects of passive (odor exposure, habituation) and active (odor reinforced with reward) olfactory learning on granule cell neurogenesis, apical and basal dendritic spines of adult-born granule cells and mitral cell activity and physiology. The authors find/claim that passive learning increased neurogenesis and consequently, increased inhibitory drive onto mitral cells (assessed using ChR2 stimulation of adultborn GCs) and decreased odor-evoked activity of the mitral cells. In contrast, the authors show that active learning produces a decrease in apical dendritic spines without affecting basal spines. Interestingly, the decrease in apical dendritic spine density was reversed with restoration of discrimination levels to baseline over time. Furthermore, odor-evoked activity in MCs was increased and ChR2 light evoked IPSC frequency of the mitral cells was decreased.

This is an interesting study that begins to delineate how different kinds of learning affect GC-MC connectivity. Because passive and active learning may affect inputs and outputs of adult born and developmentally generated GCs and mitral cells (along with physiological properties of mitral cells), it is not clear how adultborn GCs are solely driving changes in inhibition onto mitral cells under these different learning conditions.

For example, does blockade of adult GC neurogenesis under active or passive learning conditions (over the learning period) eliminate the reported effects on mitral cells?

1) It is not clear why Mann-Whitney was used over a one-way ANOVA for comparisons of apical or basal dendritic spine density.

2) Figure 1: Effect on basal spine density appears to be driven by one datapoint. Does basal spine density return to baseline after 42 days?

3) How do authors control for number of cells expressing ChR2 virus or normalize counts to interpret the ChR2 stimulation data (number of spines vs. number of cells)? Are equivalent numbers of cells infected in both groups? The increase in survival induced by learning should bias the number of Chr2 expressing GCs in OB.

4) Following reversal of changes in apical spine density (but with extra adultborn GCs still present), is the effect of ChR2 stimulation on MCs lost?

5) Although the authors examine activity of mitral cells (sparseness), evidence examining population based coding or input-output transformations is critically needed to justify the use of term pattern separation or interpret the data within this framework.

[Editors’ note: what now follows is the decision letter after the authors submitted for further consideration.]

Thank you for resubmitting your work entitled "Opposite regulation of inhibition by adult-born granule cells in response to implicit versus explicit olfactory learning" for further consideration at *eLife*. Your revised article has been favorably evaluated by Gary Westbrook (Senior Editor), a Reviewing editor (Naoshige Uchida), and three reviewers.

This is a resubmission in which the authors examined the effects of two types of olfactory learning on granule cell neurogenesis, spine density of adult-born granule cells (GCs) and mitral cell (MC) activity and physiology. They make various interesting findings. All the reviewers thought that the manuscript has greatly improved, but still raised some relatively minor concerns. We believe these points can be addressed without additional experiments, by revising the manuscript including adding discussions on some caveats or future directions. Please see below the individual points raised by the reviewers.

*Reviewer #1:*

This is a much improved paper that shows a striking difference in inhibition provided by adult-born GCs depending on the type of learning.

Is it possible that some part of the character of the effect is because of the odor similarity? Might it look different if the odors were easier to discriminate/less overlapping? I suggest the authors qualify the results under the class of fine odor discrimination. A nice follow-up study (not for this paper) would be to compare in each condition (implicit vs explicit), or at least the explicit condition, the effect of discrimination difficulty.

*Reviewer #2:*

I recommend the manuscript for publication.

Please discount the significance of the Daroles et al. study as evidence supporting a role for an increase in spine density in implicit learning. This is because FMRPcKO mice show an elevation in spine density at baseline (Figure 3) and therefore, it is not clear whether the failure to increase spine density further following learning or the elevation prior to learning is the culpable factor. Additionally, FMRP has numerous functions within neurons, that when disrupted, may be responsible for behavioral phenotype in implicit learning.

The authors acknowledge "However, while these evidences favor a prominent role of adult born neurons in shaping mitral cell activity in learning, the role of pre-existing interneurons in mitral cell response plasticity remains to be investigated". However, this possibility is absent from discussion. Please address this concern in addition to acknowledging "potential changes in mitral cell properties' also as a potential mechanism.

*Reviewer #3:*

The major concerns have been addressed appropriately, including a careful point-by-point response to the reviewers' concerns. In particular the statistics have been improved both in terms of n and of the tests, and the figures also are way more convincing. In my view the manuscript is now acceptable for publication, upon minor changes.

---

## [Author Response]

[Editors’ note: the author responses to the first round of peer review follow.]

Essential revisions:

1) All the reviewers raised their concerns regarding statistical analyses. Most importantly, the authors perform multiple pair-wise comparisons. In such cases, a significance criterion should be corrected for multiple comparisons (e.g. by using Bonferroni correction). When appropriate, a one-way ANOVA (analysis of variance) followed by post-hoc pair-wise comparisons is preferred. Many conclusions do not appear to hold after these corrections. This would mean that some conclusions need to be revised or the authors need to perform additional experiments to increase the sample size. More detailed criticisms and suggestions can be found in the reviewers' comments appended below.

We would like to thank the referees and the editor for their comments and suggestions on statistics that greatly enhance the strength of our conclusions. We have entirely revised the statistical analysis and added new data in response to reviewer’s comments. Please see detailed replies below.

2) Some of the statistical significance appear to be due to some outlier points in the data (e.g. reviewer 1, point #1; reviewer 2, point #2). The authors need to consider whether the results hold even removing these points.

The new analysis performed with larger samples confirmed the main results. See responses to reviewer 1 and 2.

To make these results more reliable, we expect that a larger sample size is required to draw statistically-sound conclusions. The reviewers also thought that the conclusion that adult-born neurons are mediating the enhanced inhibition onto MCs requires more work:

We added new data (new animals, new neurons). See table below:

Added n=total n=**BrdU**
PC26n=animalsCond26Non-Enr26Enr04**BrdU/Zif268**
PC27n=animalsCond27Non-Enr27Enr16**Morpho Apical**
PC856/5n=fragments/animalsCond0106/4Non-Enr448/4Enr030/4**Morpho Basal**
PC043/4n=fragments/animalsCond037/4Non-Enr1859/7Enr1248/7**sIPSC Frequency**
PC23/346/5n=cells/animalsCond28/134/3Non-Enr24/348/6Enr23/350/6**sIPSC Amplitude**
PC23/346/5n=cells/animalsCond28/134/3Non-Enr24/346/6Enr23/348/6

Table A: List of additional data

3) How do authors control for number of cells expressing ChR2 virus or normalize counts to interpret the ChR2 stimulation data (number of spines vs. number of cells)? Are equivalent numbers of cells infected in both groups? The increase in survival induced by learning should bias the number of Chr2 expressing GCs in OB.

In line with the reviewer’s comment and according to our BrdU counts (Figure 1 and Figure 2), we agree that enrichment or conditioning should result in an increased number of ChR2+ cells (we did not count these cells on the slices used for electrophysiology because the thickness of the sections (400 µm) makes it difficult in our view to provide reliable cell count). This leads to the conclusion that the increased inhibition on M/T cells following implicit learning could be accounted for by an increased number of adult-born cells. This issue is now discussed in the manuscript (Discussion section).

However, this does not explain how the very same number ChR2 GCs turns out to produce decreased inhibition on M/T cells in animal subjected to explicit learning. This is why we think that learning induces additional plasticity (such as changes in apical spine density onto M/T cells).

4) Following reversal of changes in apical spine density (but with extra adult-born GCs still present), is the effect of ChR2 stimulation on MCs lost?

Extra adult-born GCs expressing BrdU are no longer present 42 days after explicit learning as shown in a previous publication from our lab (Sultan et al., 2010). We now added new data to the present paper showing that extra adult-born GCs are no longer present 42 days after implicit learning, at a time where changes in apical spine density and behavioral discrimination are no longer observed (new Figure 1—figure supplement 2).

5) The link between behavior (higher discrimination) and the changes at the level of neurogenesis, morphology and physiology remain largely unclear. We understand that this study is not designed to examine the mechanisms that connect these changes with behavior, but the reviewers found some of the discussions very loose. We would like the authors to state these links more carefully, so that the manuscript becomes more scholarly. For example, the authors mention "pattern separation" multiple times and discuss that increased or decreased inhibition may be causally related pattern separations, and in turn, increased discriminability. However, whether pattern separation causally underlies increased discrimination or whether changes in inhibitory tones can lead to pattern separations remain totally unknown (also see point #4 by reviewer 3). The present study as well as existing literature does not specify specific mechanisms underlying increased discriminability after implicit or explicit olfactory learning.

In response to this comment, we have focused and rewritten the Discussion section.

We suggest that differences specifically in apical dendrite synapses onto granule cells create more or less feedback inhibition, which leads to specific changes in M/T responses to a learned odorant. After implicit learning, overall inhibition would be increased, learning to sharper odor representations in the OB. In contrast, after explicit learning, inhibition in response to the conditioned odor would be decreased, leading to stronger odor responses to the conditioned odor.

Reviewer #1:

In this manuscript, the authors describe bidirectional changes in the spine density of olfactory bulb adult-born granule cells (aGCs), that depend on learning paradigms which the authors have established previously.The used paradigms appear subtle and physiological, since they rely on few odor exposures. Nevertheless, the changes can be observed in electrophysiological recordings in brain slices even though the relation of the investigated mitral cells to odor-activated glomeruli is not known. Thus, the phenomena might reflect a general response of an otherwise deprived olfactory system.

The reviewer rightly points out that relation of the mitral cells recorded in the electrophysiological experiments to the odor-activated glomeruli is unknown. However, most odorants are documented to activate a rather large number of glomeruli (from 15 to 35 glomeruli (Vincis et al., 2012). Previous studies in our lab have shown that Limonene (and other odorants) evokes broad activation of the granule cell layer as reported using Zif268 expression (Mandairon et al., 2008; Moreno et al., 2014). Similarly, the pattern of addition of new neurons to the olfactory bulb show some degree of odor/spatial specificity but nevertheless also covers distributed regions of the olfactory bulb (Mandairon et al., 2006, Alonso et al., 2006, Sultan et al., 2010, Moreno et al., 2012). We thus agree that there is indeed a broad effect of the learning paradigms we used on the bulbar network that could be underlined by broadly distributed odor-responding cells. We also acknowledge that olfactory deprivation of laboratory mice may enhance the contrast between the learning condition and controls.

My general criticism is that there are counterintuitive results that would require stronger statistics to be firmly established. Therefore, the authors should increase sample size, try to sample cells more specifically (see below) and thoroughly discuss their results, possibly requiring a longer format.

As reported above we have substantially increased the number of animals/neurons for most experiments and performed the required better statistical methods (see Table A above).

1) Statistical significance appears to be an issue with some results – would unilateral naris occlusion be feasible, which might allow for pairwise comparisons across hemibulbs? Or a higher number of animals?

Because of contralateral projections between olfactory bulbs via the anterior olfactory nucleus, and the fact that learning after unilateral occlusion transfers to the other side (Guthrie et al., 1990), we would not be confident that the occluded side of the brain would be a good control. Instead, we have performed additional experiments to increase animal/neuron numbers (see Table A above).

While the increased spine density (Figure 1) looks fine, all the other changes (Figure 1, Figure 2) show highly overlapping distributions, perhaps because some MCs in the sample might not have been involved in limonene-evoked bulbar activation. Would it be possible to preferentially record from bulbar parts with strong limonene activation? Or even better fluorescently label MC/GCs expressing IEGs and record from them?

Statistical analysis is revised and performed on the new data sets. All morphological data (including those of Figure 1, Figure 1 and Figure 2) have been re analyzed using non parametric Kruskall-Wallis Anova (Statistica) including the four groups (Enr, Non Enr, PC and Cond) and pair-wise comparisons are done by FDR-corrected permutation tests (R). Please also note that data points have been separated on the figures to make their distribution more visible (see also Source data files including all experimental values submitted with the manuscript). The new analysis confirmed the main findings.

As stated above, the odor responding adult-born cells are broadly distributed in the OB and represent an average of ~60% of total number of BrdU labelled cells (BrdU/Zif268 co expressing cells, Figure 1 and Figure 2) and we agree that this heterogeneity may increase data variability and thereby reduce the chance to reveal an effect of learning rather than the opposite.

Finally, the ultimate experimental way to address this issue could be fluorescently labeled MC or GCs expressing IEG as suggested by the reviewer. However, to our knowledge and appreciation, in the available inducible systems, tagging is irreversible and the time window allowing expression of the IEG (several hours) is still too long to enable specific labelling of the cells responding to the odorant stimulation of interest and not to other olfactory cues. Thus, even though these designs have proven highly valuable tools for investigating memory processes, they seem less appropriate for tracking cells responding directly to sensory stimuli in physiological environment (Reijmers et al., 2007, Guenthner, 2013, Denny et al., 2014).

There is a substantial amount of eIPSC recordings with high frequency in the 30-60 Hz range in PC condition (Figure 2). This range is not observed in Figure 1, even not in the enriched case. Were the high frequency responses recorded in slices from the same animal?

For PC (Figure 2), eIPSCs were recorded from 20 cells in 2 different animals. Among the 7 values above 30 Hz, 2 were recorded from animal 1 and 5 from animal 2. For information, 19 cells from 2 animals were recorded for Cond.

Pre-light data have been added to the figure (new Figure 1—figure supplement 3 and Figure 2—figure supplement 3). In order to take into account the factors “light” (pre and post light), “group of learning” (implicit versus explicit) and “learning” (Cond or Enr versus PC or Non-Enr), we normalized the data (ln(x+1)) and performed a 3-factor Anova followed by paired when appropriate or unpaired t-tests.

To strengthen the analysis further, the percentage of “connected” mitral cells, ie cells significantly responding to the light stimulation, has been recalculated based on a statistical analysis of the change in IPSC frequency between pre and post light stimulation (rather than according to an arbitrary threshold in our previous analysis). The latter analysis is now presented in new Figure 1 and Figure 2.

Strikingly, the very same issue arises also for the sIPSC frequency and amplitude (Figure 2) – for both there are several data points in the high range for PC that are not observed in non-enr or enr conditions.

Regarding sIPSC frequency (Figure 2), we have added new animals and recording (see Table A). Total numbers of cells/animals per group now amount to: Non Enr 48/6), Enr 50/6, PC 46/5, Cond 34/3. New data are shown on new Figure 2, (PC and Cond) and Figure 1 (enr and non–enr).

Statistical analysis now also includes FDR-corrected permutation tests in which 1000 000 artificial groups were tested. The analysis yield a level of significance at p=0.06 (after correction) for the differences in sIPSCs frequency between Cond and PC and between enr and non-enr. Regarding amplitudes, the new analysis confirmed a lower sIPSC amplitude in Cond versus PC. See new Figure 1 and Figure 2.

If we now doubt the PC data for this reason, what about the statistical comparison between non-enr and cond? Unfortunately, this comparison is missing from Table 1 – by eye between Figure 1 and Figure 2 almost certainly there is no difference for the apical spine density. Which casts doubt on the central finding of reduced connectivity – at least anatomically.

Regarding apical spine density, we performed a Kruskall Wallis Anova including the 4 groups and FDR-corrected permutation tests on the new data set. Importantly, Cond showed a significantly lower apical spine density than non-enr (p=0.013). See new Figure 1 and Figure 2 and new Table 1 in the manuscript that now includes all comparisons.

2) What is the effect of the increase in number of adult born GCs also in the associative learning – how exactly can an addition of interneurons reduce inhibitory drive? As mentioned above, the decrease in spine numbers on the apical dendrite is barely significant. Nevertheless, there is a striking overall decrease in inhibitory drive of mitral cells, especially substantially reduced sIPSC frequency and amplitude (please label the examples from Figure 2 and Figure 1 top in the distributions shown below). If we believe these data (but see above), what is going on here? Synaptic plasticity – reduction in release probability and/or quantal size at the GC-MC contacts – also for preexisting GCs? Were the lost contacts located close to the MC somata and thus the efficiency is strongly reduced?

The apical spine density in the conditioned group is significantly lower than in the pseudoconditionned group, p=0.0017 (Anova including the 4 groups and FDR-corrected permutation tests).

We added representative images (new Figure 1 and Figure 2).

The new adult-born granule cells replace older ones (Imayoshi et al., 2008). Our hypothesis, now developed in the discussion is that “replacing pre-existing granule cells by new ones with fewer synaptic contacts with mitral cells (in conditioned animals) would result in a global pool of granule cells delivering less local inhibition in response to the conditioned odor. In contrast, replacing granule cells by new cells making more local and global synaptic contacts with mitral cells (enriched animals) would result in a shift toward more inhibition in the network. This is consistent with experimental observations.”

3) Missing of important reference: Zhang, Huang and Hu (2016): increased spine density on GC dendrites following odor enrichment in Xenopus tadpoles.

This reference has been added

Reviewer #2:

This report by Mandairon and colleagues is a significant new look on the function of adult born GABAergic neurons in the olfactory bulb. The community as a whole thinks in two ways about the types of learning that occur with passive exposure and conditioning. On the one hand, we all know that these are different types of learning and on the other we often refer to them as if they are the same (under the umbrella term of Learning). Similarly, for learning-based effects on the survival and integration of new granule cells in the OB. This paper shows that the ability to discriminate hard-to-discriminate odors after the two types of learning may depend in opposite fashion on neurogenesis. For passive learning, it appears that adult-born GC connections with MCs are strengthened in the external plexiform layer (at the reciprocal synapse). The authors conclude that this may be related to the effect known as sparsening. For active learning, the effects are opposite. The adult born GC effects on MCs are decreased. The argument is rigorous and rational. My only major concern regards the many pairwise statistical tests (see below). The results reported here, if improved statistical methods confirm them, may constitute one of the most significant findings in the field of olfactory learning as it relates to neurogenesis.

We thank the referee for his/her very positive comment on the manuscript. We reply below to his/her specific comments.

1) There are many more implied comparisons than reported in Table 1. With 4 groups of mice, this makes 6 total comparisons for each factor (cond vs. PC, cond vs. enr, cond vs. non-enr, PC vs enr, PC vs non-enr, enr vs non-enr). With the multiple comparisons within each factor, the threshold for significance should be 0.00833. There is a pretty good argument that all of the factors should be combined in a single analysis, which then multiplies the comparisons and lowers the threshold value, or at least that like-kind factors be grouped (spine density apical/basal as one group, e/sIPSP frequency as another). This would make the p threshold smaller by a factor of 2 at least. Please provide justification in the methods for not performing the multiple comparisons adjustment or do the analysis with the corrected p thresholds. In addition to lowering the threshold, a more conservative analysis might provide additional insight.

Anova including the 4 groups (Cond, PC, Enr, non-Enr) followed by Bonferroni post hoc tests was performed for analyzing the BrdU, BrdU/Zif268, Tbx21/Zif268. For apical spine density, basal spine density and sEPSC frequency and amplitude, non-parametric Anovas followed by FDR-corrected permutation tests were performed. This was done on the new data sets including new animals and/or cell counting and recording.

For eIPSCs, in order to take into account the factors “light” (pre and post light), “group of learning” (implicit versus explicit) and “learning” (Cond or Enr versus PC or Non-Enr), we normalized the data (ln(x+1)) and performed a 3-factor Anova followed by paired or selected unpaired t-tests when appropriate. In addition, individual cell analysis of the effect of light was performed by comparing the number pre and post light IPSC across repetitions of light stimulation by unilateral Chi squared tests. These changes in statistical analyses are now included in the Materials and methods and Results sections.

2) This is not a disagreement but rather a different interpretation. It is the adult born cells that show the effects reported (and assuming that the earlier born cells do not). In the passive condition, the result is that new GCs inhibit MCs more at the apical dendrites where primary processing happens. Also, we know that these new cells are relatively specific to the enriched odors (previous work by the first and second to last authors). Is it possible that these new connections serve to help the mice ignore the conditioned stimuli and that the enhanced discrimination ability is a side-effect of this now-active ignorance? On the other hand, in the case of active learning there are more cells born and integrated (perhaps associated with the learned odors). Because these cells survive, one assumes that the inputs from higher order areas are strong. The signal is amplified, so one might expect that the now meaning-based odor perception is accomplished in the AON or PC or both, rather than in the OB.

We thank to the referee for his/her stimulating comment. Although the proposed hypothesis cannot be ruled out based on our data, and if we understood well, we could argue that during enrichment, we previously reported that the locus Coeruleus is solicited (Rey et al., 2011, Moreno et al., 2012) suggesting that daily olfactory stimulation triggers arousal, a notion is rather opposite to ignoring stimuli. Now, whether “active” ignorance is a component of arousal remains an open question. Regarding associative learning, we agree that the AON or PC may play an important part in perceiving the meaning odorants. However, the OB is still crucial to implement this amplification as suggested by the present data, and to maintain it since forgetting correlates to the loss of adult-born cells (Sultan et al., 2010, Sultan et al., 2011 and present data).

Reviewer #3:

Mandairon and colleagues examined the effects of passive (odor exposure, habituation) and active (odor reinforced with reward) olfactory learning on granule cell neurogenesis, apical and basal dendritic spines of adult-born granule cells and mitral cell activity and physiology. The authors find/claim that passive learning increased neurogenesis and consequently, increased inhibitory drive onto mitral cells (assessed using ChR2 stimulation of adultborn GCs) and decreased odor-evoked activity of the mitral cells. In contrast, the authors show that active learning produces a decrease in apical dendritic spines without affecting basal spines. Interestingly, the decrease in apical dendritic spine density was reversed with restoration of discrimination levels to baseline over time. Furthermore, odor-evoked activity in MCs was increased and ChR2 light evoked IPSC frequency of the mitral cells was decreased.This is an interesting study that begins to delineate how different kinds of learning affect GC-MC connectivity. Because passive and active learning may affect inputs and outputs of adult born and developmentally generated GCs and mitral cells (along with physiological properties of mitral cells), it is not clear how adultborn GCs are solely driving changes in inhibition onto mitral cells under these different learning conditions.For example, does blockade of adult GC neurogenesis under active or passive learning conditions (over the learning period) eliminate the reported effects on mitral cells?

From our previous work on perceptual learning, we know that blocking neurogenesis abolishes the increase in paired-pulse inhibition of mitral cells and GAD expression in the OB (Moreno et al., 2009). In addition, blocking neurogenesis (Moreno et al., 2009) or structural plasticity in adultborn cell (Daroles et al., 2015) prevents implicit learning. Regarding associative learning, we have shown that blocking neurogenesis during learning altered recall of the discrimination task (5 days post learning, a delay similar to that in the present study). Finally, in the present work, we show that the morphological changes in spine density on granule cells return to pre-learning level after one month, consistent with the loss of the behavioral response. However, while these evidences favor a prominent role of adult born neurons in shaping mitral cell activity in learning, the role of pre-existing interneurons in mitral cell response plasticity remains to be investigated.

1) It is not clear why Mann-Whitney was used over a one-way ANOVA for comparisons of apical or basal dendritic spine density.

The statistical analysis was entirely revised. Anova followed by FDR-corrected permutation tests were performed. (See above, reply to editor comments and to referees #1 & #2.)

2) Figure 1: Effect on basal spine density appears to be driven by one datapoint.

We have added new data in this data set (more neurons analyzed) and statistical analysis was performed by non-parametric Anova followed by FDR-corrected permutation tests:

-basal spine density differed between Enr and non-enr (Figure 1) (Kruskall Wallis H(3, 187)=20.15, p<0.0001, enr versus non enr p=0.0015)

-PC did not differed from Cond (PC versus Cond p=0.09).

Does basal spine density return to baseline after 42 days?

Data regarding basal spine density after 42 days has been added to the manuscript. Basal spine density does return to pre learning value (new graph in Figure 1—figure supplement 2).

3) How do authors control for number of cells expressing ChR2 virus or normalize counts to interpret the ChR2 stimulation data (number of spines vs. number of cells)? Are equivalent numbers of cells infected in both groups? The increase in survival induced by learning should bias the number of Chr2 expressing GCs in OB.

In line with the reviewer’s comment and according to our BrdU counts (Figure 1 and Figure 2), we agree that enrichment or conditioning should result in an increased number of ChR2+ cells (we did not count these cells on the slices used for electrophysiology because the thickness of the sections (400 µm) makes it difficult in our view to provide reliable cell count). This leads to the conclusion that the increased inhibition on M/T cells following implicit learning could be accounted for by an increased number of adult-born cells. This issue is now discussed in the manuscript (Discussion section).

However, this does not explain how the very same number ChR2 GCs turns out to produce decreased inhibition on M/T cells in animal subjected to explicit learning. This is why we think that learning induces additional plasticity (such as changes in apical spine density onto M/T cells).

This view is supported by our previous work (Daroles et al., 2016) showing that blockade of the increase in spine density is sufficient to prevent perceptive learning.

4) Following reversal of changes in apical spine density (but with extra adultborn GCs still present), is the effect of ChR2 stimulation on MCs lost?

Extra adult-born GCs expressing BrdU are no longer present 42 days after explicit learning as shown in a previous publication from our lab (Sultan et al., 2010). We now added new data to the present paper showing that extra adult-born GCs are no longer present 42 days after implicit learning, at a time where changes in apical spine density and behavioral discrimination are no longer observed (new Figure 2).

5) Although the authors examine activity of mitral cells (sparseness), evidence examining population based coding or input-output transformations is critically needed to justify the use of term pattern separation or interpret the data within this framework.

We agree that pattern separation strictly refers to the process of input-output transformation and we have modified the discussion accordingly.

[Editors' note: the author responses to the re-review follow.]

Reviewer #1:

This is a much improved paper that shows a striking difference in inhibition provided by adult-born GCs depending on the type of learning.Is it possible that some part of the character of the effect is because of the odor similarity? Might it look different if the odors were easier to discriminate/less overlapping? I suggest the authors qualify the results under the class of fine odor discrimination. A nice follow-up study (not for this paper) would be to compare in each condition (implicit vs explicit), or at least the explicit condition, the effect of discrimination difficulty.

Done.

Reviewer #2:

I recommend the manuscript for publication.Please discount the significance of the Daroles et al. study as evidence supporting a role for an increase in spine density in implicit learning. This is because FMRPcKO mice show an elevation in spine density at baseline (Figure 3) and therefore, it is not clear whether the failure to increase spine density further following learning or the elevation prior to learning is the culpable factor. Additionally, FMRP has numerous functions within neurons, that when disrupted, may be responsible for behavioral phenotype in implicit learning.

This part of the sentence has been removed from the Discussion section.

The authors acknowledge "However, while these evidences favor a prominent role of adult born neurons in shaping mitral cell activity in learning, the role of pre-existing interneurons in mitral cell response plasticity remains to be investigated". However, this possibility is absent from discussion. Please address this concern in addition to acknowledging "potential changes in mitral cell properties' also as a potential mechanism.

This sentence has been added to the Discussion section.